# Nucleobindin-2/Nesfatin-1—A New Cancer Related Molecule?

**DOI:** 10.3390/ijms22158313

**Published:** 2021-08-02

**Authors:** Alicja M. Kmiecik, Piotr Dzięgiel, Marzenna Podhorska-Okołów

**Affiliations:** 1Division of Histology and Embryology, Department of Human Morphology and Embryology, Wroclaw Medical University, 50-368 Wroclaw, Poland; piotr.dziegiel@umed.wroc.pl; 2Department of Physiotherapy, University School of Physical Education, 51-612 Wroclaw, Poland; 3Division of Ultrastructure Research, Wroclaw Medical University, 50-368 Wroclaw, Poland; marzenna.podhorska-okolow@umed.wroc.pl

**Keywords:** NUCB2, NESF-1, nucleobindin-2, nesfatin-1, cancer

## Abstract

Cancer is a heterogeneous disease, and even tumors with similar clinicopathological characteristics show different biology, behavior, and treatment responses. As a result, there is an urgent need to define new prognostic and predictive markers to make treatment options more personalized. According to the latest findings, nucleobindin-2/nesfatin-1 (NUCB2/NESF-1) is an important factor in cancer development and progression. Nucleobindin-2 is a precursor protein of nesfatin-1. As NUCB2 and nesfatin-1 are colocalized in each tissue, their expression is often analyzed together as NUCB2. The metabolic function of NUCB2/NESF-1 is related to food intake, glucose metabolism, and the regulation of immune, cardiovascular and endocrine systems. Recently, it has been demonstrated that high expression of NUCB2/NESF-1 is associated with poor outcomes and promotes cell proliferation, migration, and invasion in, e.g., breast, colon, prostate, endometrial, thyroid, bladder cancers, or glioblastoma. Interestingly, nesfatin-1 is also considered an inhibitor of the proliferation of human adrenocortical carcinoma and ovarian epithelial carcinoma cells. These conflicting results make NUCB2/NESF-1 an interesting target of study in the context of cancer progression. The present review is the first to describe NUCB2/NESF-1 as a new prognostic and predictive marker in cancers.

## 1. Introduction

Nucleobindin 2 (NUCB2) was first described in 1994 in KM3 acute lymphoblastic leukemia cell line as a DNA binding/EF-hand/acidic-amino acid-rich protein [1,2]. It has been extensively studied since Oh-I et al. identified nesftain-1 as a NUCB2 cleavage product [3]. Several reports indicate that NUCB2/NESF-1 is also expressed in many organs and tissues (e.g., in the stomach, pancreas, heart, reproductive organs, and adipose tissue) [4,5,6,7,8]. Nucleobindin-2/Nnesfatin-1 is also a secretory component of various body fluids, including saliva, synovial fluid, milk, and plasma or serum [9,10,11,12,13]. As NUCB2 and nesfatin-1 are colocalized, these two names are used interchangeably. The NUCB2/NESF-1 gene in humans is located on chromosome 11 and consists of 14 exons spanning 54 785 nucleotides [14]. Nucleobindin-2 is a 396-amino acid protein preceded by a 24-amino acid signal peptide [15]. The protein is proteolytically cleaved by the prohormone convertase. As a result, N-terminal nesfatin-1, nesfatin-2, and the C–terminal nesfatin-3 are formed [16]. The functions of nesfatin-2 and the C–terminal nesfatin-3 have not been understood yet [17]. Nucleobindin-2 has characteristic functional domains, such as a signal peptide, a Leu ⁄ Ile rich region, two Ca2+ binding EF-hand domains separated by an acidic amino acid-rich region, and a leucine zipper. Therefore it may play a role in many cellular processes [18,19] (Figure 1). Despite the increasing knowledge about the expression and regulation of NUCB2/NESF-1, its role in physiology and pathology is still poorly understood. 

## 2. NUCB2/NESF-1 in Physiology

This protein is involved in the regulation of many intracellular processes. Its metabolic function includes food intake, glucose metabolism, and the regulation of immune, cardiovascular, and endocrine systems [20,21,22,23]. Recently, nesfatin-1 has been identified as an anorexigenic neuropeptide that seems to play a crucial role in appetite regulation and energy homeostasis [24]. Oh-I et al. found that intracerebroventricular (IVC) injection of nesfatin-1 decreased food intake in a dose-dependent manner in rats [3]. Furthermore, lower nesfatin-1 concentrations were observed in obese children compared to healthy children [25]. It was also suggested that NUCB2/NESF-1 expressed in the parvocellular region contributed to anorexia by secreting nesfatin-1 [26,27,28]. Nesfatin-1 was reported as a suppressor of gastric emptying and an inhibitor of gastroduodenal motility in mice and rats [21,29]. The expression of NUCB2/NESF-1 was also observed in pancreatic islets, which suggests its role in glucose homeostasis [30]. Su et al. demonstrated that nesfatin-1 reduced blood glucose levels in hyperglycemic db/db mice [31]. Moreover, Gonzales et al. demonstrated that nesfatin-1 enhanced glucose-stimulated insulin secretion in mouse pancreatic β- cells [32]. Nesfatin-1 was found to be an anti-inflammatory and antiapoptotic agent in rats with traumatic brain injury [33]. It was demonstrated that the administration of nesfatin-1 30 min after head trauma could significantly suppress inflammatory-related proteins such as tumor necrosis factor-alpha (TNF-α), IL-1β (interleukin-1 β), and IL-6 (interleukin-6). Moreover, it was also shown that nesfatin-1 reduced the activity of caspase-3 and the number of apoptotic neuronal cells in traumatic rat brain tissue. It is known that NUCB2/NESF-1 is implicated in the inflammatory response by its involvement in the tumor necrosis factor/tumor necrosis factor receptor 1 (TNF/TNFR1) signaling. Islam et al. revealed that tumor necrosis factor receptor 1 release required the calcium-dependent formation of NUCB2/NESF-1 and aminopeptidase regulator of TNFR1 shedding (ARTS-1) to promote exosome-like vesicle release in human vascular endothelial cells [34]. Feijóo-Bandín et al. showed that mouse and human cardiomyocytes could synthesize and secrete nesfatin-1, which helps regulate glucose metabolism in the hearts of humans and experimental animals [35]. Additionally, the plasma concentration of nesfatin-1 in patients with acute myocardial infarction (AMI) was lower than in the control group, which showed nesfatin-1 to be a protective agent against AMI [36]. On the other hand, it was also found that the infusion of nesfatin-1 to the cerebral cortex and subcutaneous tissue resulted in elevated blood pressure, while intravenous infusion led to vasoconstriction [37]. The role of nesfatin-1 in the cardiovascular system is still poorly understood. There is some evidence that NUCB2/NESF-1 is related to female pubertal maturation. Garcia Galano found that during the female pubertal transition, NUCB2/NESF-1 mRNA and protein levels were significantly increased in the hypothalamus [38]. Intracerebroventricular (ICV) injections of nesfatin-1 induced increased circulating levels of gonadotropin-releasing hormone, luteinizing hormone (LH), and follicle-stimulating hormone (FSH) [39]. Expression of NUCB2/NESF-1 was detected in Leydig cells in rodents and humans. During the pubertal transition, NUCB2/NESF-1 protein level in these cells was significantly increased [38]. To conclude, recent studies have identified nuclebindin-2/nesfatin-1 as a pleiotropic peptide with many physiological functions. Recently, the expression of NUCB2/NESF-1 has been linked to tumor development and metastasis. However, the exact role of NUCB2/NESF-1 in human malignancies is still poorly understood (Figure 2). The current review is the first to present NUCB2/NESF-1 as a potential new prognostic or predictive marker in cancers.

## 3. NUCB2—A Predictive/Prognostic Biomarker in Cancers?

Cancer has become a socioeconomic problem of the contemporary world, and it still remains one of the major healthcare challenges. In recent years, identification and the characteristics of new cancer biomarkers have become major targets for cancer studies. Biomarkers are useful for diagnosis, monitoring disease progression, predicting disease recurrence, and therapeutic treatment efficiency [40]. Although many new cancer-related molecules have been recently discovered, there is an urgent need to define new prognostic and predictive markers to make treatment options more personalized and effective. Here we summarize available data about NUCB2/NESF-1 expression in relation to clinicopathological properties in different cancer types. The summary below gives evidence that suggests that NUCB2/NESF-1 is a potential new biomarker in different cancer types and introduces it as a new area for cancer research.

### 3.1. Breast Cancer

We evaluated the expression of NUCB2 in breast cancer tissues in our current research (data not published yet). The expression of NUCB2/NESF-1 was localized in the cytoplasm of breast cancer cells (Figure 3A). No expression was found in the breast cancer cell nucleus or the tumor stroma (Figure 3A). The presence of NUCB2/NESF-1 detected by immunohistochemistry (IHC) was higher in breast carcinoma compared to benign changes in the glandular tissue in the breast (mastopathy; control) (Figure 3B).

Suzuki et al. were the first to identify NUCB2/NESF-1 as an estrogen receptor-associated protein. They found NUCB2/NESF-1 to be a gene associated with the recurrence of estrogen receptor-positive breast carcinoma patients (n = 5) [41]. Furthermore, it was found that a higher NUCB2/NESF-1 level was observed in tumors with estrogen receptor expression and was positively associated with lymph node metastasis. It is well established that breast cancers that express functional estrogen receptor (ER) exhibit a less aggressive metastatic phenotype than ER-negative cases. Interestingly, it was shown that NUCB2/NESF-1 status was significantly associated with an increased risk of recurrence. Multivariate analysis demonstrated that NUCB2/NESF-1 was an independent prognostic factor for disease-free survival. No significant association was found between the presence of NUCB2/NESF-1 and age, menopausal status, clinical stage, tumor size, histological grade, mitotic score, progesterone receptor (PR) status, human epidermal growth factor receptor 2 (HER2) status, or Ki-67 [41]. Zeng et al. confirmed a positive correlation between NUCB2/NESF-1 expression and the ER status.

Additionally, they found a significant positive correlation between nodal metastasis and the clinical stage. Moreover, it was demonstrated that breast cancer patients with high NUCB2/NESF-1 expression had a significantly poorer overall survival (OS) [42]. Interestingly, the survival analysis with an online analysis tool on 2032 breast cancer cases indicated that a good prognostic effect of high NUCB2 expression was related to longer OS [43]. These conflicting results suggest an important role of NUCB2/NESF-1 in breast cancer progression and highlight a need for further investigation in this area. 

### 3.2. Colon Cancer

Kan et al. were the first to detect NUCB2/NESF-1 expression in colon cancer. The immunofluorescence analysis showed that the expression of NUCB2/NESF-1 was higher compared to non-tumor regions. Additionally, they assessed serum nesfatin-1 levels in colon cancer patients. The obtained results revealed no difference in nesfatin-1 concentration between healthy donors and colon cancer patients [44]. In 2018, Xie et al. found that NUCB2/NESF-1 mRNA was upregulated in colorectal cancer (CRC) tissues compared to the noncancerous tissue obtained from the same patient. Additionally, immunohistochemistry showed the expression of NUCB2/NESF-1 in 251 colon cancer patients in relation to clinicopathological properties. The protein was predominantly expressed in the cytoplasm and much less in the cancer cell membrane. The results also indicated a positive correlation between NUCB2/NESF-1 expression and lymph node metastasis and the TNM stage. Patients with lymph node metastasis showed a higher expression of NUCB2/NESF-1 compared to those without lymph node metastasis (49.5%, vs. 36.6%, *p* = 0.043). Moreover, TNM stage III-IV patients had significantly higher NUCB2/NESF-1 expression compared to TNM stage I-II patients (50.9% vs. 35.0%). No significant association was found between NUCB2/NESF-1 expression and disease-free survival or overall survival in colon cancer patients [45]. To conclude, NUCB2/NESF-1 is suggested to be associated with aggressive progression in colorectal cancer.

### 3.3. Bladder Cancer

Liu et al. assessed NUCB2/NESF-1 expression in 115 cases of bladder cancer patients in relation to clinicopathological tumor characteristics. Their study showed that high expression of NUCB2/NESF-1 was associated with distant metastasis and vascular invasion. Patients with high NUCB2/NESF-1 levels had poor overall survival and progression-free survival rates. No association was found between NUCB2/NESF-1 expression and age, tumor size, lymph node metastasis, tumor grade, or recurrence [46]. Additionally, Man Cho et al. presented that high expression of NUCB2/NESF-1 appeared relevant to aggressive clinicopathological features such as tumor grade, TNM stage, tumor size, and tumor number. According to results obtained from multivariate analysis, it was concluded that NUCB2/NESF-1 was the overall survival (OS) independent prognostic factor of bladder cancer [47]. No association was found between NUCB2/NESF-1 and gender or age. The above data confirm that NUCB2/NESF-1 may be a new biomarker of bladder cancer progression.

### 3.4. Prostate Cancer

The NUCB2/NESF-1 mRNA level was analyzed in 180 pairs of prostate cancer tissues and the corresponding noncancerous tissue. The analyses revealed that the expression of NUCB2/NESF-1 was significantly higher in cancer cells compared to the noncancerous control. The upregulation of NUCB2/NESF-1 mRNA in prostate cancer was correlated with higher Gleason scores (*p* < 0000.1), higher levels of preoperative prostate-specific antigen (PSA) (*p* = 0.0004), positive lymph node metastasis (*p* = 0.022) and positive angiolymphatic invasion (*p* = 0.0004). No relationship was found between NUCB2/NESF-1 and age, seminal vesicle invasion, pathological stage, or surgical margin status. Additionally, patients with low NUCB2/NESF-1 mRNA levels had significantly longer biochemical recurrence-free survival (BCR-free—the survival time of a person with prostate cancer during which a biochemical marker—PSA does not rise or rises very little) time after radical prostatectomy compared to patients with high NUCB2/NESF-1 mRNA levels. Moreover, multivariate analysis demonstrated that a high NUCB2/NESF-1 mRNA level was an independent predictor of shorter BCR-free survival [48]. In other studies, the same research team showed that high NUCB2/NESF-1 mRNA expression was related to the poor overall survival of patients with prostate cancer. In addition, multivariate Cox analysis indicated that NUCB2/NESF-1 mRNA was an independent prognostic factor for the overall survival of prostate cancer patients [48]. Immunohistochemistry evaluation revealed that NUCB2/NESF-1 expression was significantly higher in prostate cancer compared to benign prostatic hyperplasia (*p* < 0000.1) [49]. Furthermore, it was also shown that NUCB2/NESF-1 protein expression was significantly associated with seminal vesicle invasion, higher levels of preoperative PSA, positive lymph node metastasis, positive angiolymphatic invasion, biochemical recurrence, and higher Gleason scores. Multivariate Cox regression analysis revealed that a high NUCB2/NESF-1 protein expression level was an independent prognostic factor for overall survival and BCR-free survival of patients with prostate cancer [14]. To conclude, the above evidence suggests that NUCB2/NESF-1 may be an important biomarker in prostate cancer patients. 

### 3.5. Endometrial Cancer

Takagi et al. examined the clinical significance of NUCB2/NESF-1 in 87 cases of endometrial carcinoma (EC). Immunoreactivity of NUCB2/NESF-1 was detected in 22% of endometrial carcinomas and significantly higher than non-neoplastic endometrial glands. This protein was positively correlated with the Ki-67 marker of proliferation. No association was found between NUCB2/NESF-1 expression and other clinicopathological parameters. The NUCB2/NESF-1 status was significantly associated with an increased risk of recurrence (*p* = 0.004). Moreover, univariate analysis revealed that NUCB2/NESF-1 status was an independent prognostic factor for disease-free survival and cancer-specific survival [50]. Markowska et al. found a high cytoplasmic expression of nesfatin-1 in 53.1% of patients with endometrial cancer. No relationship was found between the intensity of nesfatin-1 expression and the clinical stage of the disease in the whole patient cohort. Additionally, no relationship was detected between the level of NUCB2/NESF-1 expression and histological grade of differentiation (*p* = 0.3145) in all patients. Interestingly, there are two types of EC that differ in terms of etiology, biology, and clinical course, i.e., type I EC (endometrioid adenocarcinoma) and type II EC (non-endometrioid adenocarcinoma). The study results demonstrated that in type I EC, nesfatin-1 expression was significantly higher in G1 than G2 and G3 (in total; G1—68.97% compared to G2 and G3—50.4%, *p* = 0.0487) [51]. To conclude, the presented results suggest that the expression of NUCB2/NESF-1 may be significant for the prognosis in EC patients. However, its role is still poorly understood. 

### 3.6. Gastric Cancer

The immunohistochemistry analysis revealed that NUCB2/NESF-1 in gastric cancer cells was predominantly localized in the nuclei and its expression was higher in tumor tissues than in the adjacent normal tissues. The expression of NUCB2/NESF-1 was significantly associated with the tumor depth, lymph node metastasis, lymphatic invasion, venous invasion, and clinical stage. Additionally, univariate analysis showed that a significant relationship was found between NUCB2/NESF-1 expression and unfavorable progression-free survival and overall survival. Moreover, multivariate analysis revealed that NUC2/NESF-1 was an independent predictor of progression-free survival [52]. To conclude, NUCB2/NESF-1 might be used as a new biomarker and a potential therapeutic target for gastric cancer.

### 3.7. Papillary Thyroid Cancer

The immunohistochemistry evaluation of the presence of NUCB2/NESF-1 in 155 cases of papillary thyroid cancer revealed nucleus and cytoplasmic expression. Normal tissue adjacent to cancer was not stained. The results demonstrated that NUCB2/NESF-1 was significantly associated with extrathyroidal extension, TNM stage, and tumor size (*p* < 0.05). No relationship was found between NUCB2/NESF-1 and age, gender, or lymph node metastasis. [53]. Bearing in mind the above, it was speculated that NUCB2/NESF-1 could be a tumor promoter in papillary thyroid cancer. 

### 3.8. Renal Cell Carcinoma

Using immunohistochemistry, Qi et al. analyzed the expression level of NUCB2/NESF-1 in 188 clear cell renal cell carcinoma (ccRCC) samples and adjacent noncancerous tissues. The expression of the protein was significantly higher in tumor tissues compared to the control. Expression of NUCB2/NESF-1 was associated with the T stage and the presence of metastasis (*p* = 0.0006 and <0.001). No significant relationship was found between NUCB2/NESF-1 protein expression and age, tumor size, Furhman grade, or tumor necrosis. Patients with high NUCB2/NESF-1 expression had a shorter overall survival rate. Moreover, univariate and multivariate Cox regression models revealed that NUCB2/NESF-1 was an independent prognostic factor for overall survival [54]. In turn, Fu et al. showed that NUCB2/NESF-1 was positively correlated with the Fuhrman grade (*p* < 0.002) and the presence of necrosis. Moreover, it was also demonstrated that patients with high NUCB2/NESF-1 expression had a shorter cancer-specific survival rate. In addition, NUCB2/NESF-1 seems to be an independent prognostic factor for cancer-specific survival in ccRCC [55].

In 2019, Wei et al. revealed that NUCB2/NESF-1 was highly expressed in the renal cell carcinoma (RCC) tissues compared with the adjacent noncancerous. The NUCB2/NESF-1 protein expression was not significantly associated with age, gender, tumor stage, Fuhrman grade, or tumor recurrence (*p* > 0.05). However, NUCB2/NESF-1 protein expression was associated with perinephric tissue invasion, cancerous thrombus, and distant metastases (*p* = 0.044 and 0.002, respectively) [56]. Tao et al. confirmed that the level of NUCB-2/NESF-1 in patients with renal cell cancer (n = 68) was significantly increased compared to the normal control patients (n = 10) [57]. To conclude, NUCB-2/NESF-1 was identified as a valid marker associated with tumorigenesis and progression of renal cancer.

### 3.9. Glioblastoma

Liu et al. found that the mRNA expression level of NUCB2/NESF-1 in glioblastoma was significantly higher than in normal tissues (*p* < 0.05, n = 163) [58]. The IHC staining showed that NUCB2/NESF-1 was predominantly expressed in the nucleus and was highly overexpressed in glioblastoma compared to the adjacent tissue [58]. Moreover, it was noticed that the high expression of NUCB2/NESF-1 was related to the recurrence. No association was observed between NUCB2/NESF-1 expression and other clinical features, including age, gender, tumor lateralization, or IDH1 mutations. The above data demonstrated that NUCB2/NESF-1 was associated with a poor prognosis of glioblastoma and may be used as a potential therapeutic target for this disease in the future. 

## 4. Regulation of NUCB2 Expression in Cancers

In general, the expression of NUCB2 in cancers is significantly higher compared to normal control. While NUCB2 has been intensively studied in recent years, little is known about the promoter of this gene. Suzuki et al. revealed that the *NUCB2* gene contained a functional estrogen response element (ERE) in the promoter region [41]. This element is a specific DNA sequence found in the regulatory regions of estrogen-responsive genes [59]. The influence of estradiol treatment on MCF-7 cells was analyzed. It was demonstrated that NUCB2/NESF-1 expression was upregulated by estradiol in estrogen receptor-positive MCF-7 cells [41]. 

Zhang et al. indicated that knockout of KLF4 (Krüppel-like factor 4) in Mel-RM melanoma cells resulted in suppression of NUCB2. KLF4 is a zinc finger-type transcription factor that usually binds GC-rich elements of the promoters. Two putative KLF4 binding sites on the NUCB2 promoter were found. Moreover, IHC studies revealed that the expression of NUCB2 was positively correlated with KLF4 in melanoma tissues. It is well known that KLF4 plays an important role in tumorigenesis. Still, it can function either as a tumor suppressor or as an oncogene, depending on the type of the tissue or cancer and even on the tumor stage [60]. This feature may be crucial for understanding NUCB2 regulation in different cancer types.

MicroRNAs (miRNAs) are endogenous small non-coding RNAs that function in the regulation of gene expression. In cancer cells, dysregulated miRNAs have been shown to influence the hallmarks of cancer, including sustaining proliferative signaling, evading growth suppressors, resisting cell death, activating invasion and metastasis, and inducing angiogenesis [61,62]. Depending on the circumstances, miRNA may function either as a tumor suppressor or as an oncogene. Huo et al. were the first to indicate the involvement of miRNA in NUCB2 regulation in cancer [63]. They revealed that NUCB2 was a target for miRNA-335-5p. MicroRNA-335-5p is known as a potential suppressor of metastasis and invasion in gastric or thyroid cancer cells. The current study has shown that MicroRNA-335-5p is sponged with lncRNA FTX (long non-coding RNA) in the lung adenocarcinoma cell line. lncRNAs can affect cancer progression by silencing microRNAs (miRNAs) and releasing messenger RNAs [64]. This phenomenon resulted in a higher expression of NUCB2 and promotion of cancer cell progression. Similar research was conducted by Xin et al. They demonstrated that NUCB2 was a target for miR-149. In renal cancer cells, miR-149 is sponged by circular RNA circ_00150, which resulted in the NUCB2 increase [65]. Circular RNAs (circRNAs) are other RNA molecules, which play critical roles in the development and progression of cancers [66]. Li et al. confirmed the involvement of miRNA in NUCB2 regulation. NUCB2 was revealed to be a target gene of miR-30a-5p in nasopharyngeal carcinoma (NPC). Moreover, it was shown that miR-30a-5p was downregulated in NPC tissues and cells. In vitro experiments confirmed that miR-30a-5p suppressed the proliferation, migration, and invasion of NPC cells via targeting NUCB2. Li et al. concluded that miR-30a-5p modulated NPC progression by targeting NUCB2 and suggested that these findings lay a foundation for exploring the clinical treatment of NPC [67]. Bearing in mind the above evidence, regulation of NUCB2 in cancer is an intricate process that requires further investigation.

## 5. NUCB2 in Cancer Proliferation, Apoptosis, Migration and Invasion

Proliferation is one of the most important characteristics of cancer. Some proteins are overexpressed in cancer cells and indicated as proliferation markers, which makes them useful in cancer diagnostics. Among the best-evaluated cell proliferation molecules, we classified, e.g., Ki-67, proliferating cell nuclear antigen (PCNA), or minichromosome maintenance (MCM) proteins. Although the proteins have been extensively described in the literature, they also have some limitations when used in the diagnostic process [68]. As a result, there is a need to identify new proliferation-related molecules in cancers. In vitro studies showed that NUCB2/NESF-1 knockdown with siRNA or shRNA in breast cancer (MCF-7, SKBR-3), bladder cancer (T24, 5637), glioblastoma (U251, U87), endometrial cancer (Ishikawa and Sawano), and thyroid cancer (TPC-I, KI) cell lines resulted in the inhibition of cell proliferation [41,47,50,53,58]. Moreover, proliferation-related proteins (Ki-67 and PCNA) were decreased in NUCB2/NESF-1 shRNA-transfected thyroid cancer cells compared to the control. To evaluate the role of NUCB2/NESF-1 related to the growth and metastasis of glioblastoma, NUCB2/NESF-1-silenced cells were injected into nude mice. The findings indicated that the NUCB2/NESF-1 ablation group’s tumor volume was significantly smaller than the control [58]. Additionally, NUCB2/NESF-1 knockdown in the 786-O renal cancer cell line inhibited cell proliferation by arresting the cell cycle at the S phase. Moreover, reduced tumor volume and growth rate were observed in NUCB2/NESF-1-knockdown renal cancer cells in the mice model compared to that in the negative control group [57]. Interestingly, no changes were found in the proliferation properties of NUCB2-knockdown colon cancer cells [44]. Surprisingly, Ramanjaneya et al. found that treatment of H295R adrenocortical cells with recombinant nesfatin-1 resulted in a decreased proliferative capacity of the cells [69]. They put a hypothesis that NUCB2/NESF-1 might be a therapeutic target for adrenal cancer. Similar conclusions were made by Xu et al., who evaluated the treatment of HO-8910 ovarian cancer cells with recombinant human nesfatin-1. It was revealed that nesftain-1 decreased cell proliferation in ovarian cancer in vitro [70]. Surprisingly, treatment of the endometrial cancer cell line (Ishikawa) with recombinant nesfatin-1 promoted cell proliferation [50]. Dysregulation of cell proliferation is known to be related to the mTOR signaling cascade. Takagi et al. revealed that NUCB2/NESF-1 increased mTOR phosphorylation, which resulted in the intense proliferation of the endometrial cancer cell line [50]. Contradictory findings were presented by Xu et al., who reported that NUCB2/NESF-1 decreased mTOR phosphorylation and acted as a tumor suppressor in ovarian cancer [70]. Similarly, Ramanjaneya et al. indicated that NCB2/NESF-1 decreased ERK1/2 phosphorylation, which resulted in decreased proliferation in H295R adrenocortical carcinoma cells [69]. Bearing in mind the above, we may conclude that the role of NUCB2/NESF-1 in cancers is variable and tissue-specific. Additionally, Xu et al. concluded that nesfatin-1 could inhibit the proliferation in human ovarian epithelial carcinoma cell line HO-8910 by inducing apoptosis via the mTOR and RhoA/ROCK signaling pathway [70]. Rho Kinases (ROCKs) are known to be modulators of cell survival and apoptosis. It was reported that NUCB2/nesfatin-1 treatment evoked a marked activation of RhoA, which enhanced apoptosis and inhibited proliferation.

The balance between cell proliferation and apoptosis is crucial for normal development and homeostasis in adults [71]. An imbalance between these two processes causes cancer. Apoptosis is a well-known mechanism of programmed cell death, which plays a crucial role in development and homeostasis. The loss of apoptotic control allows cancer cells to survive and favors tumor progression. One of the major challenges in cancer treatment is identifying factors that terminate the uncontrolled growth of cancer cells [72]. The effects of nesfatin-1 on apoptosis of H295R adrenocortical cells using the DNA fragmentation assay were assessed. The study found that nesfatin-1 induced a concentration-dependent increase in apoptosis of H295R cells compared to control cells. Moreover, the analyses of pro-and-anti-apoptotic gene expression following nesfatin-1 treatment in H295R cells were performed. A significant increase was found in the proapoptotic Bax (*p* < 0.05), while a significant decrease was reported in antiapoptotic BCL-XL (*p* < 0.01) and BCL-2 (*p* < 0.05) mRNA expression following nesfatin-1 stimulation [69]. The study may contribute to cancer prevention and therapy.

However, Xu et al. found that the downregulation of NUCB2/NESF-1 in the 786-O renal cancer cell line resulted in an increased apoptosis rate, making NUCB2 an antiapoptotic factor. The above results indicate that the function of NUCB2 may vary among tissues, and further investigation is needed to understand the molecular mechanism underlying these diverse roles of NUCB2/NESF-1 [73].

Cell migration and invasion are important steps in cancer metastasis, resulting in disseminating primary tumor cells to distant organs [74]. Metastasis is a major cause of death in cancer patients. There is a constant need to identify potential molecular therapeutic targets which are involved in migration, invasion, and metastasis. In vitro studies revealed that NUCB2/NESF-1 knockdown in MCF-7 (ER+) and SKBR-3 (ER-) inhibited cell migration and invasion properties [41]. Interestingly, in 2013, Gest et al. evaluated breast cancer aggressiveness with the MCF-7 cell line. They concluded that the MCF-7 cells did not migrate or invade. As a result, these cells seem to be a controversial model to investigate the role of NUCB2 in breast cancer invasion [75]. The suppression of NUCB2/NESF-1 with siRNA inhibited migration and invasion in the colon, renal, Ishikawa, and Sawano endometrial cancer cells [46,50]. Additionally, treatment of Ishikawa cells with recombinant nesfatin-1 promoted cell migration. Lower invasion and migration properties were found in NUCB2/NESF-1-knockdown glioblastoma cells. Moreover, it was shown that the occurrence of lung metastasis was clearly decreased in mice injected with NUCB2/NESF-1-knockdowned U251 cells compared to the control [58]. Additionally, decreased invasion ability of NUCB2/NESF-1-silenced TPC-I and KI thyroid cancer cell lines was observed compared to the control [53]. Metalloproteinases (MMPs), such as MMP-2 and MMP-9 gelatinases, have the matrix-degrading ability and are well known for cell motility and invasion [76]. The study showed that knockdown of NUCB2/NESF-1 in thyroid cancer cells decreased invasion-related proteins (MMP-2 and MMP-9) [53]. The study results also demonstrated that NUCB2/NESF-1 shRNA-silenced bladder cancer cell lines had a lower migration and invasion ability than controls. In addition, it was also found that NUCB2/NESF-1-knockdowned cells had a lower expression of MMP-2 and MMP9, which explains the above findings. Moreover, in vitro studies with NUCB2/NESF-1-knockdowned cells injected BALB/c mice revealed lung metastases only in the control group [46]. 

To summarize, NUCB2 seems to be an important factor in cancer cell migration and invasion and may affect the expression of MMP-2 and MMP-9. The mechanism underlying this observation is unknown. 

It has been well established that cell invasion during cancer progression may be dependent on the acquisition of epithelial-mesenchymal transition (EMT) features [77]. This transition is a major process in embryonic development. It has been revealed that EMT is related to cancer progression. EMT is a complex process in which epithelial cells lose cell-cell adhesion structures and polarity, obtain cell motility and acquire the characteristics of invasive mesenchymal cells [78]. This is a key step in the metastatic cascade. Recent studies show the role of NUCB2 in the EMT of cancer cells. Suppression of NUCB2/NESF-1 in colon cancer cells resulted in morphological changes in the clones, which suggested that NUCB2/NESF-1 was involved in the EMT phenomenon. Twist and Slug are important transcription factors, which regulate EMT phenotypes. The levels of TWIST and Slug were significantly lower in NUCB2/NESF-1-knockdowned cells, while E-cadherin, β-catenin, and Claudin-3 (epithelial markers) increased in these cells compared to the controls. Kan et al. showed that the mRNA level of ZEB-1, a regulator of EMT genes, was associated with NUCB-2 expression. In addition, they revealed that ZEB-1 had a critical role in NUCB-2- mediated migration, invasion and EMT pathways in colon cancer [46]. The ZEB1 expression is also regulated through mTOR and AMPK signaling pathways. It was proposed that AMPK might regulate ZEB1 expression through mTOR and TORC1 signaling pathways and enhance EMT. NUCB2/NESF-1 knockout in the SK-RC-52 renal cancer cell line affected EMT-related proteins such as E-cadherin, β-catenin, Slug, and Twist. The results demonstrated that knockdown of NUCB2/NESF-1 expression in SK-RC-52 cells increased the phosphorylation of AMPK and decreased mTOR phosphorylation [57]. As a result, ZEB1 activity was significantly inhibited. 

The above study confirmed that NUCB-2 might promote EMT via the AMPK/TORC1/ZEB1 pathway in cancer. In addition, a positive correlation was found between the expression of NUCB2/NESF-1 and EMT-related genes such as desmoplakin *(DSP)*, Integrin Subunit Alpha V *(ITGAV)*, metalloproteinase 3 (*MMP3)*, tetraspanin 13 ( *TSPAN13),* and Cadherin 2 (*CDH2)* in endometrial cancer cells [52].

To summarize, NUCB-2 seems to play an important role in cancer progression (Figure 4). Although the available data shows it as a prooncogenic factor related to proliferation, migration, invasion, EMT, or evading apoptosis, it is crucial to notice that single reports are presenting nesfatin-1 as a tumor suppressor (e.g., in adrenocortical and ovarian cancer cells), which inhibited cell proliferation and increased apoptosis. The expression and the role of NUCB2/NESF-1 in adrenocortical and ovarian tumors tissues has not been evaluated yet. There are still different types of cancer (e.g., pancreatic cancer) that have not been analyzed in the context of NUCB2 expression. Pancreatic ductal adenocarcinoma (PDA) is a lethal malignancy with a dismal 5-year survival rate of only 10%, making it one of the most aggressive cancers worldwide [79]. Recent studies succeeded in isolating different pancreatic cell types from mouse and human pancreas [80,81]. Due to the difficulties in obtaining pancreatic cancer tissue for research, cell lines may be useful as a model to indicate the role of NUCB2 in this cancer type. The NUCB2 protein has several functional domains and proteolytic derivatives (i.e., nesfatin-1, nesftain-2, nesftain-3). Although huge progress related to the NUCB2/nesfatin-1 function in cells has been recently achieved, the mechanisms of action of the protein (such as regulation of NUCB2 processing and release of nesftain-1, nesftain-2, and nesfatin-3) have not been revealed yet. We suspect that the role of NUCB2/NESF-1 may be different in various tissues and cell types.

## 6. Conclusions

The first study on nucleobidin-2/nesfatin-1 function in cancer was made in 2012 in breast cancer patients [41]. Since then, there have been single reports on NUCB2/NESF-1 in the most common human cancers, i.e., breast, colon, gastric, bladder, ovarian, endometrial, prostate, papillary thyroid cancers, renal cell carcinoma, and glioblastoma (Table 1) [41,46,47,48,53]. In general, the expression of NUCB2/NESF-1 is associated with cancer progression. It was a poor prognostic factor in breast cancer, clear cell renal carcinoma, gastric, prostate, endometrial, and bladder cancer [42,47,50,52]. In vitro studies revealed that NUCB2/NESF-1 increases invasion, migration, and proliferation in the colon, bladder, papillary thyroid, endometrial, renal, breast cancer cells, and glioblastoma [41,46,50,58]. Interestingly, nesfatin-1 is also considered an inhibitor of the proliferation of human adrenocortical and ovarian epithelial carcinoma cells [69,70]. Moreover, treatment of adrenocortical and ovarian epithelial carcinoma cells with nesfatin-1 resulted in increased apoptosis. On the other hand, NUCB2/NESF-1 knockdown in renal cancer cells increased cell apoptosis. Currently, little is known about the NUCB2/NESF-1 mechanism of action in cancer cells. Recent studies have indicated that NUCB2/NESF-1 enhanced migration, invasion, and EMT properties of colon and renal cancer cells through AMK/TOC1/ZEB1 pathway activation [44,57]. Takagi et al. demonstrated that the mTOR protein was significantly phosphorylated after nesfatin-1 treatment of the endometrial carcinoma cell line, which resulted in enhanced cell proliferation and migration properties [50]. However, Xu et al. revealed that nesfatin-1 decreased mTOR phosphorylation, which resulted in the inhibition of proliferation in human ovarian epithelial carcinoma cell line HO-8910 through inducing apoptosis via the RhoA/ROCK signaling pathway [70]. These studies suggest the NUCB2/NESF-1 may affect mTOR pathways in different tissues, and these signaling pathways induce different functions. 

It is known that NUCB2/NESF-1 is involved in the inflammatory response by its involvement in the TNF/TNFR1 signaling pathway [34]. Extracellular tumor necrosis factor receptors (TNFR) function as TNF-binding proteins that modulate the activity of TNF [82], which can induce such diverse effects as apoptosis, necrosis, angiogenesis, immune cell activation, differentiation, and cell migration [83]. These processes are crucial for tumor progression. NUCB2/NESF-1 interacts with ART-1, an integral membrane protein associated with TNFR1, and promotes its release to the extracellular compartment [84]. Therefore, by interacting with different partners, NUCB2/NESF-1 may be involved in tumor development. NUCB2/NESF-1 is considered both cancer suppression and progression factors. The protein has several functional domains and proteolytic derivatives, and hence, its role may be different among tissues and cell types. Moreover, a putative receptor for nesftain-1 has not been identified yet. To identify nesfatin-1 specific receptor and its mechanism of action, further studies are warranted. Identification, localization, and modulation of the NESF-1 receptor will be crucial for a better understanding of nesfatin-1 signaling pathways.

In conclusion, NUCB2/NESF-1 might be used as a new biomarker and a potential therapeutic target for different cancer types in the future.

## Figures and Tables

**Figure 1 ijms-22-08313-f001:**
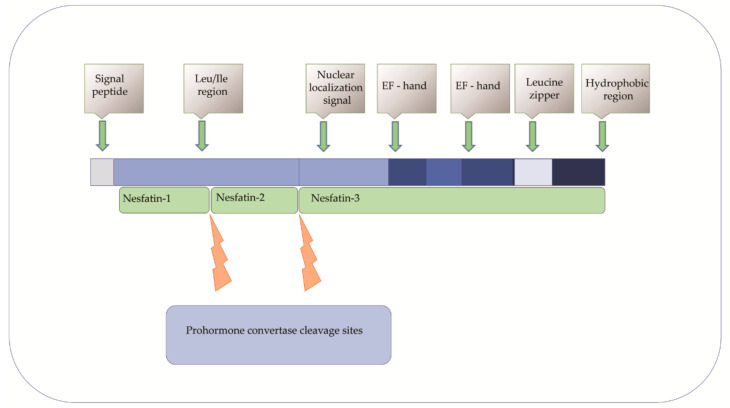
Structural domains of NUCB2 and its processing to protein derivatives, i.e., nesfatin-1, nesfatin-2, and nesfatin-3.

**Figure 2 ijms-22-08313-f002:**
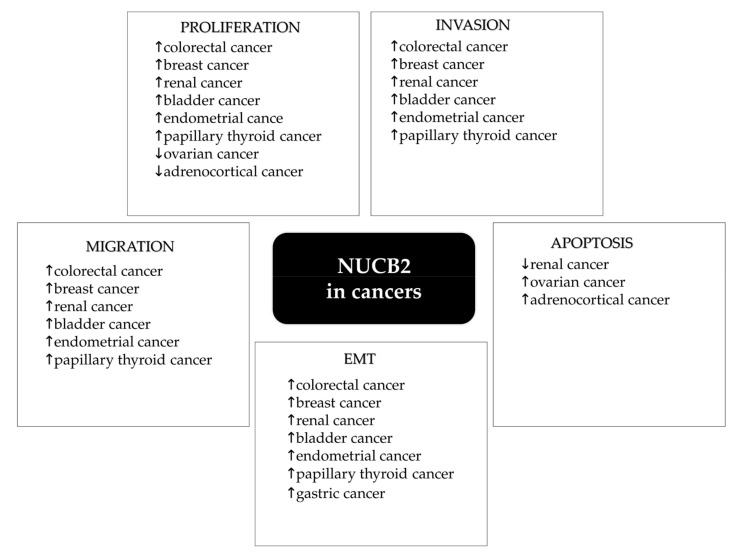
Role of NUCB2/NESF-1 in human tumor progression.

**Figure 3 ijms-22-08313-f003:**
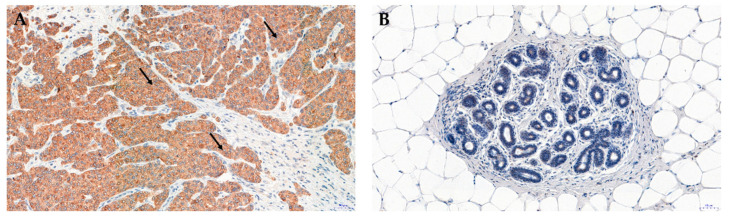
Expression of NUCB2/NESF-1 in invasive ductal breast carcinoma (**A**) and mastopathy (**B**) NUCB2/NESF-1 was located in the cytoplasm of breast cancer cells (black arrows) (**A**). Original magnification: ×200; for the immunohistochemical demonstration of NUCB2/NESF-1, Autostainer Link 48 (Dako, Glostrup, Denmark) with the visualization system of EnVisione FLEX, High pH (Dako) was used. NUCB2 was detected by incubation of deparaffinized sections with primary anti-nucleobindin 2 antibodies (rabbit polyclonal antibody, Novus Biologicals, Littleton, CO, USA).

**Figure 4 ijms-22-08313-f004:**
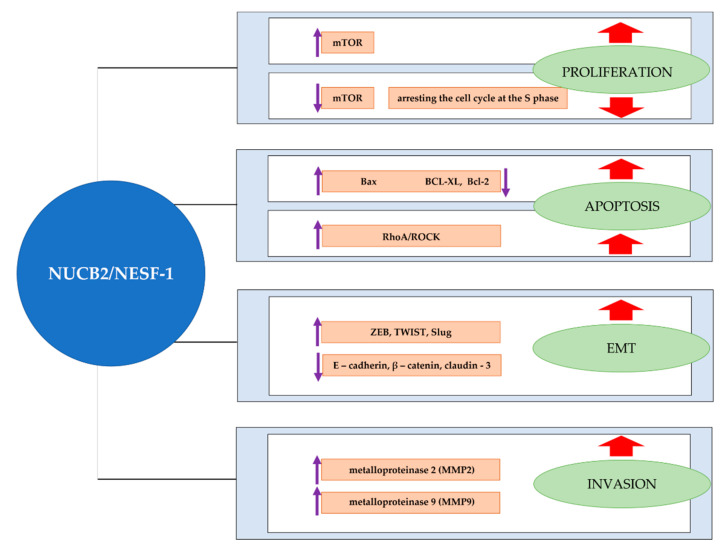
Involvement of NUCB2/NESF-1 in cancer-related processes. NUCB2/NESF-1 may stimulate cancer cell proliferation and apoptosis via the mTOR signaling cascade. NUCB2/NESF-1 also enhances EMT and cancer cell invasion by regulating gene expression associated with these processes, such as TWIST, Slug, and MMP-2 or MMP-9. On the other hand, NUCB2/NESF-1 plays the role of a tumor suppressor promoting apoptosis and in cancer cells via mTOR and RhoA/ROCK pathways.

**Table 1 ijms-22-08313-t001:** Expression and function of NUCB2/NESF-1 in different cancer types.

Cancer Type	Evaluation Method	NUCB2/NESF-1 Expression/Function	Ref.
Breast cancer	IHC	NUCB2/NESF-1 was positively associated with the ER status of breast carcinoma patients, lymph node metastasis, clinical stage, and an increased risk of recurrence; NUCB2/NESF-1 was an independent prognostic factor for disease-free survival.	[41,42]
	In vitro	Inhibition of NUCB2/NESF-1 expression in MCF-7 and SKBR-3 resulted in decreased proliferation, invasion, and migration properties; NUCB2 expression was upregulated by estradiol in ER-positive MCF-7 cells.	
Colon cancer	IF	The expression of NUCB2/NESF-1 in cancer was higher than in non-tumor regions; NUCB2/NESF-1 was predominantly expressed in the cytoplasm and much less in the cancer cell membrane. The results also indicated a positive correlation between NUCB2/NESF-1 expression and lymph node metastasis and the TNM stage.	[45]
	ELISA	There was no difference in serum nesfatin-1 concentration between healthy donors and colon cancer patients.	
	In vitro	NUCB2/NESF-1 enhanced EMT, migration, and invasion in colon cancer cells.	
Bladder cancer	IHC	High expression of NUCB2/NESF-1 was associated with distant metastasis and vascular invasion. Patients with high NUCB2/NESF-1 had poor overall survival and progression-free survival rates	[46]
	In vitro	Suppression of NUCB2/NESF-1 using shRNA in T24 and 5637 bladder cancer cell lines resulted in the inhibition of cell proliferation, migration, and invasion abilities compared to the control. NUCB2/NESF-1 knockdowned cells had a lower expression of MMP2 and MMP9.	
	In vivo	The growth of tumors from NUCB2/NESF-1knockdowned cells in BALB/c mice was slower compared to the control, and lung metastases were observed only in the control group.	
Ovarian cancer	In vitro		
		Administration of recombinant human nesfatin-1 resulted in enhanced apoptosis and decreased ovarian cancer cell proliferation.	[70]
Prostate cancer	real time RT—PCR, IHC	Expression of NUCB2/NESF-1 was significantly higher in cancer cells than noncancerous control and correlated with higher Gleason scores, higher levels of preoperative PSA, positive lymph node metastasis and positive angiolymphatic invasion. High NUCB2/NESF-1 mRNA level was an independent predictor of shorter BCR-free survival. Multivariate Cox analysis indicated that NUCB2/NESF-1 mRNA was an independent prognostic factor for the overall survival of prostate cancer patients.	[47,48,49]
Gastric cancer	IHC	NUCB2 was localized in the nuclei, and its expression was higher in the tumor than in the adjacent normal tissues. NUCB2 was significantly associated with tumor depth, lymph node metastasis, lymphatic invasion, venous invasion, and clinical stage. NUC2/NESF-1 was an independent predictor of progression-free survival. A positive correlation was found between the expression of NUCB2/NESF-1 and EMT-related genes such as *DSP*, *ITGAV*, *MMP3*, *TSPAN13,* and *CDH2*.	[52]
Papillary thyroid cancer	IHC	Papillary thyroid cancer showed nucleus and cytoplasmic expression with no staining in the normal tissue adjacent to cancer. NUCB2/NESF-1 was significantly associated with extrathyroidal extension, TNM stage, and tumor size.	[53]
	In vitro	Inhibition of NUCB2/NESF-1 expression in TPC-I and KI thyroid cell lines resulted in decreased proliferation, invasion, and migration properties and decreased expression of invasion-related proteins (MMP-2 and MMP).	
Renal cell carcinoma	IHC	The expression of NUCB2/NESF-1 was significantly higher in the tumor compared to the control. NUCB2/NESF-1 expression was associated with the T stage and the presence of metastasis. High NUCB2/NESF-1 expression was associated with a shorter overall survival rate. NUCB2/NESF-1 was an independent prognostic factor for overall survival. NUCB2 was positively correlated with the Fuhrman grade (*p* < 0.002) and the presence of necrosis.	[55,56,57]
	In vitro	Inhibition of NUCB2/NESF-1 expression in the 786-O renal cancer cell line resulted in an increased apoptosis rate and a decreased invasion rate. NUCB2/NESF-1 knockout in the SK-RC-52 cell line inhibited migration, invasion, and affected EMT-related proteins	
Adreno- cortical cancer	In vitro	Treatment of H295R cells with nesfatin-1resulted in a decreased proliferative capacity. Nesfatin-1 induced a concentration-dependent increase in apoptosis of H295R cells compared to control cells	[69]
Glioblastoma	Real-time RT- PCR, IHC	mRNA expression level of NUCB2/NESF-1 in glioblastoma was significantly higher than in normal tissues. NUCB2/NESF-1 was predominantly expressed in the nucleus and was highly overexpressed in glioblastoma compared to the adjacent tissue. High expression of NUCB2/NESF-1 was related to recurrence.	[58]
	In vitro	Inhibition of NUCB2/NESF-1 expression in U251 and U87 glioblastoma cell lines resulted in decreased proliferation, invasion, and migration properties	
	In vivo	The tumor volume of the NUCB2/NESF-1 ablation group was significantly smaller than the control; the occurrence of lung metastasis was decreased in mice injected with NUCB2/NESF-1 knockdowned U251 cells compared to the control.	
Endometrial cancer	IHC	NUCB2/NESF-1 was positively correlated with the Ki-67 marker of proliferation. No association was found between NUCB2/NESF-1 expression and other clinicopathological parameters. NUCB2/NESF-1 status was significantly associated with an increased risk of recurrence (*p* = 0.004). NUCB2/NESF-1 status was an independent prognostic factor for disease-free survival and cancer-specific survival.	[50]
	In vitro	Knockdown of NUCB2/NESF-1 using a specific siRNA significantly inhibited Ishikawa and Sawano endometrial cancer cell proliferation and migration. Treatment of Ishikawa cells with recombinant nesfatin-1 also promoted cell proliferation and migration.	

## Data Availability

Not applicable.

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
