# Peer review of "Nucleobindin-2/Nesfatin-1—A New Cancer Related Molecule?"

_ijms, 2021, doi:10.3390/ijms22158313_

Round 1
Reviewer 1 Report
I have already reviewed this paper in the past. At that time, authors made some corrections into the review. Now, they are resubmitting a significantly modified version. After these two rounds of modifications, I feel that the review is now more interesting.
I have one comment that authors should include in the text:
- Pancreatic cancer is one of the most aggressive cancers worldwide with few therapeutic options, yet the role of NUCB2 in this cancer has not been investigated. Recent studies succeeded to isolate different pancreatic cell types from mouse and human pancreata. This may allow the study of NUCB2 in pancreas malignancy at the single-cell level in the future. Related to this discussion, authors might also cite these studies:
- PMID: 33782087 ;
- PMID: 29535635 ;
- PMID: 33212097;
- PMID: 27667667
Reviewer 2 Report
The revised manuscript meets the previous suggestions. The manuscript is suggested to be accepted in the present form.
Author Response
We would like to thank the Reviewer for his/her time and a constructive review of our study.
This manuscript is a resubmission of an earlier submission. The following is a list of the peer review reports and author responses from that submission.
Round 1
Reviewer 1 Report
In this manuscript, the authors summarized the current findings on Nesfatin-1 and Nucelobindin-2 on the function of cancer progression and predicts NESF-1/NUCB2 as a novel target on cancer studies. Overall, the review includes most recent publications and studies, and it's function in different cancer types are highlighted with references. The structure of the manuscript is well-organized. The manuscript is important to highlight the critical role of NESF-1/NUCB2 and bring new ideas in cancer studies and treatments. Thus, the study is suggested to be published.
Author Response
We would like to thank the Reviewer for his/her time and a constructive review of our study.
Reviewer 2 Report
The manuscript by Alicja M. Kmiecik et al. reviewed the biological function and potential clinical relevance of Nucleobindin-2/Nesfatin-1, which are relatively new molecules in cancer biology. The paper has a clear outline and well-organized supporting evidences. The topic is potentially interesting to the readers in the biology and oncology field.
However, I feel not convinced by the evidence in the field so far that Nucleobindin-2/Nesfatin-1 merit the wide attention of readers. First, the functional link of Nucleobindin-2/Nesfatin-1 with tumorigenesis is vague, mostly based on in vitro studies, and molecular mechanism not dissected clearly. Second, although there are large amount of retrospective, small-scale clinical correlation studies suggesting Nucleobindin-2/Nesfatin-1 as potential prognostic biomarker of various cancer types, the really conclusive evidence comes from randomized, prospective clinical studies. Based on the concern above, I’m afraid I can’t recommend further consideration of this manuscript by IJMS for publication.
Minor points:
- Nomenclature of nucleobindin-2/nesfatin-1 (NUCB2/NESF-1) should be double checked throughout the paper.
- Reference format should be improved.
Author Response
Response to Reviewer 1 Comments
Point 1: The manuscript by Alicja M. Kmiecik et al. reviewed the biological function and potential clinical relevance of Nucleobindin-2/Nesfatin-1, which are relatively new molecules in cancer biology. The paper has a clear outline and well-organized supporting evidences. The topic is potentially interesting to the readers in the biology and oncology field.
However, I feel not convinced by the evidence in the field so far that Nucleobindin-2/Nesfatin-1 merit the wide attention of readers. First, the functional link of Nucleobindin-2/Nesfatin-1 with tumorigenesis is vague, mostly based on in vitro studies, and molecular mechanism not dissected clearly. Second, although there are large amount of retrospective, small-scale clinical correlation studies suggesting Nucleobindin-2/Nesfatin-1 as potential prognostic biomarker of various cancer types, the really conclusive evidence comes from randomized, prospective clinical studies. Based on the concern above, I’m afraid I can’t recommend further consideration of this manuscript by IJMS for publication.
Resposne 1:
We would like to thank the Reviewer for his/her time and a constructive review of our study.
Cancer has become a socioeconomic problem of the contemporary world and it still remains one of a major healthcare challenges. It is a heterogenous disease and even tumors with similar clinicopathological characteristics presents a different biology or treatment response. Therefore, it is important to identify new molecules associated with cancer development. Our paper is the first to summarize the current state of knowledge on the expression and role of NUCB2/NESF-1 in cancers. The first study on NUCB2/NESF-1 function in cancer was made in 2011. Since then, there have been single reports on NUCB2/NESF-1, which suggested that NUCB2/NESF-1 might be a new prognostic and/or predictive marker in different cancer types. According to the available literature data, we believe that NUCB2/NESF-1 is an important target in cancer studies. We agree that the functional link of NUCB2/NESF-1 with tumorigenesis is vague. Further studies are warranted to explain the role of NUCB2/NESF-1 in tumors and to answer the question whether the proteins are cancer-related molecules. The intentions of the authors was to indicate the role of NUCB2/NESF-1 in tumorgenesis and to introduce it as a new area for cancer research. We have followed the Reviewer’s suggestions and decided to change the title and put it as a question (by adding a question mark “?”). Therefore, the previous title Nucleobindin-2/Nesfatin-1 – a New Cancer-Related Molecule was changed to
Nucleobindin-2/Nesfatin-1 – a New Cancer-Related Molecule ?
Point 2: Nomenclature of nucleobindin-2/nesfatin-1 (NUCB2/NESF-1) should be double checked throughout the paper.
Respone 2: Nomenclature of nucleobindin-2/nesfatin-1 was corrected.
Point 3: Reference format should be improved.
Response 3: Reference format was improved.
Thank you for this minor comments.
Additionally, the English language has been verified by a medical translator and interpreter Assistant Professor Arkadiusz Badziński, Doctor of Health Sciences, University of Silesia and a native speaker. Cohesion, coherence, grammar, punctuation, spelling and medical register were preserved in accordance with the rules of the English language. The language certificate can be provided at the request of the Reviewer.
Thank you very much for your attention,
Sincerely Yours
Alicja Kmiecik
Reviewer 3 Report
In this review Kmiecik et al describe the role of NUCB2 in cancer. The review lacks dynamic discussion, it only cites selected literature without sufficient interpretation of the data. Several points should be addressed before further consideration:
- I don't see references related to a specific work done by authors regarding the role of NUCB2 in cancer. For a review paper, the authors must be experts in the field with a personal research contribution (as first or last author). I encourage authors to cite their own work in the review and discuss it more appropriately. If authors don't have any research contribution in this field, I will not accept this paper for publication
- Define in the text what is masopathy and why it is used as a control instead of normal breast tissue
- What does mean “status” when talking about ER or NUCB2 in lines 117-118? It that expression, activation, interaction? Authors should change the word “status” with more specific terms.
- Lines 125-126, what does mean MCF-7 did not show invasion ability? Is that result described in control MCF-7 cells or MCF-7 cells with NUCB2 knockdown? Also, the lines above describe that removal of NUCB2 has an effect on invasion in MCF-7 cells, this an opposite result. Authors should better discuss this part of the manuscript.
- What is exactly the expected subcellular location of NUCB2? Cytoplasm, membrane or nucleus? And how does the function of NUCB2 change depending on its subcellular location? This must be added in the text under a specific subheading (for example: “subcellular localization of NUCB2 and its related functions”). In addition, does the location and function change depending on the isoforms, nefstatin-1, 2 etc?
- Authors cited that NUCB2 as expressed in the pancreas. Pancreatic cancer is an important disease, authors should add a paragraph regarding the role of NUCB2 in pancreatic cancer
- Lines 180-186 cells were treated with recombinant nefstatin-1. How extracellular nefstatin-1 acts on the cells? Is there any specific known receptors?
- Line 196 what does biochemical recurrence-free survival stands for? This should be defined.
- Lines 233-234, what does G1, G2 and G3 stand for? Are those different stages of endometrial cancer?
- Line 369, authors said that NUCB2 exert both pro-oncogenic and tumor suppressor roles. Surprisingly, the review cited only the pro-oncogenic roles of NUCB2, authors must add significant studies reporting a tumor suppressor role for NUCB2 in cancer. This information must be added to all cancer types cited in the review.
Author Response
Response to Reviewer 3 Comments
We would like to thank the Reviewer for his/her time and a constructive review of our study.
Point 1: The review lacks dynamic discussion, it only cites selected literature without sufficient interpretation of the data.
Response 1: Discussion was extended with more aspects in the revised manuscript.
Lines: 380 – 389
Lines: 400 - 426
Point 2: I don't see references related to a specific work done by authors regarding the role of NUCB2 in cancer. For a review paper, the authors must be experts in the field with a personal research contribution (as first or last author). I encourage authors to cite their own work in the review and discuss it more appropriately. If authors don't have any research contribution in this field, I will not accept this paper for publication
Response 2:
We evaluated the role of NUCB2 in non-small-cell lung carcinoma and breast cancer cells in our current research. We received funding from the National Science Centre, Poland (project title: “Expression of NUCB2 in non-small-cell lung carcinoma (NSCLC)”, number: 2019/03/X/NZ5/00485) and from Wroclaw Medical University, Poland (project title: ”The role of NUCB2 in breast cancer progression”, number: STM.A350.20.064). The original research papers are under preparation. Moreover, we presented our results at the 53rd Symposium of the Polish Society for Histochemistry and Cytochemistry "From ultrastructure to in vivo imaging: progress in microscopical techniques". Gdańsk, 15-18 September 2019. Program, abstracts Gdańsk 2019, Polish Society for Histochemistry and Cytochemistry; Department of Histology Medical University of Gdańsk, s.142 poz.P64, 978-83-61216-07-0 (poster title: ”Expression of nucleobindin-2 in breast cancer cells”). The current review is our preliminary study to obtain the best knowledge on the role of NUCB2 in cancers. Additionally, our paper is the first to summarize the current state of knowledge on the expression and the role of NUCB2/NESF-1 in cancers. According to the available literature data, we believe that NUCB2/NESF-1 is an important target in cancer studies. The functional link of NUCB2/NESF-1 with tumorigenesis is vague. Further studies are warranted to explain the role of NUCB2/NESF-1 in tumors and to answer the question whether the proteins are cancer-related molecules. The intention of the authors was to indicate the role of NUCB2/NESF-1 in tumorigenesis and to introduce it as a new area for cancer research. Moreover, the authors have been conducting the research to identify and characterize new cancer-related molecules for many years (i.e. Kmiecik AM, Pula B, Suchanski J, Olbromski M, Gomulkiewicz A, Owczarek T, Kruczak A, Ambicka A, Rys J, Ugorski M, Podhorska-Okolow M, Dziegiel P. Metallothionein-3 Increases Triple-Negative Breast Cancer Cell Invasiveness via Induction of Metalloproteinase Expression. PLoS One. 2015 May 1;10(5):e0124865; Ratajczak-Wielgomas K, Kmiecik A, Grzegrzołka J, Piotrowska A, Gomulkiewicz A, Partynska A, Pawelczyk K, Nowinska K, Podhorska-Okolow M, Dziegiel P. Prognostic Significance of Stromal Periostin Expression in Non-Small Cell Lung Cancer. Int J Mol Sci. 2020 Sep 24;21(19):7025; Jablonska K, Nowinska K, Piotrowska A, Partynska A, Katnik E, Pawelczyk K, Kmiecik A., Glatzel-Plucinska N, Podhorska-Okolow M, Dziegiel P. Prognostic Impact of Melatonin Receptors MT1 and MT2 in Non-Small Cell Lung Cancer (NSCLC). Cancers (Basel). 2019 Jul 17;11(7):1001. doi: 10.3390/cancers11071001. PMID: 31319607; PMCID: PMC6679108.
Please find below abstract drafts of original research papers which are under preparation:
Abstract 1: Expression of NUCB2 in non-small cell lung cancer
Lung cancer is one the most frequently diagnosed types of cancer worldwide. Non-small cell lung cancer (NSCLC) accounts for 85% of all lung cancers. Despite the development of new targeted therapies, NSCLC is still one of the leading causes of cancer-related deaths. Five-year survival rate of NSCLC patients in Poland does not exceed 15%. Therefore, there is an urgent need for defining new prognostic and predictive factors to make treatment options and recommendations more personalized. Recently, it has been demonstrated that high expression of NUCB2 is associated with poor outcome and promotes cell proliferation, migration and invasion in breast, colon, prostate, endometrial, glioblastoma, thyroid or bladder cancer. Interestingly, NUCB2 is also demonstrated as an inhibitor of the proliferation of human adrenocortical carcinoma and ovarian epithelial carcinoma cells. These conflicting results point towards a possible tissue-specific regulatory function of NUCB2. This is the very first study to evaluate a clinical role of NUCB2 in NSCLC. The aim of this study was to analyze the relationship between NUCB2 expression in NSCLC and clinicopathological factors to determine its prognostic value on a large group of 775 cases of NSCLC. The obtained results revealed a statistically significant higher level of NUCB2 in the cytoplasm of NSCLC cancer cells compared to the normal lung parenchyma. The expression level of NUCB2 decreases in higher tumor sizes (T) and advanced clinical stages. Moreover, NUCB2 expression in NSCLC cells correlated weakly negatively with the expression of the Ki-67 antigen (r = -0,1477, p < 0.0001). The results of the presented studies suggested an important role of NUCB2 in NSCLC cancer transformation. However, we did not observe any relationship between the expression of NUCB2 in cancer cells and the overall survival or five-year survival rate in the analyzed cohort. In vitro studies revealed high NUCB2 mRNA and protein level in all analyzed NSCLC cells lines. The presented results suggest that NUCB2 may play a role in the malignant transformation of NSCLC and in tumor progression.
Abstract 2: Expression of nucleobindin-2/nesfatin-2 in breast cancer cells.
Introduction: NUCB2/NESF-1 is a multifunctional protein which plays an important role in food intake, energy homeostasis, insulin release, adipocyte differentiation, and the regulation of endocrine, immune and cardiovascular systems. There are several reports indicating that NUCB2/NESF-1 is expressed in numerous organs and tissues such as brain, stomach, pancreas, reproductive organs or adipose tissues. Recently, the expression of NUCB2/NESF-1 is linked to tumor development and metastasis but the exact role of NUCB2/NESF-1 in human malignancies remains unknown. We report NUCB2/NESF-1 expression and its relations to clinicopathological properties in breast cancer cells. Material and methods: Immunohistochemical reactions were conducted on 415 cases of invasive ductal carcinoma (IDC) and 36 cases of mastopathy with rabbit anti-human polyclonal antibodies against NUCB2/NESF-1. NUCB2/NESF-1 expression was evaluated using the semi-quantitative immunoreactive score (IRS) of Remmele and Stegner. The expression of NUCB2/NESF-1 was also examined at the protein level using Western blot and confocal microscopy as well as at mRNA level using real time PCR in MDA-MB-231, T47D, MCF-7 and SKBR-3 breast cancer cells line. Results: A statistically significant higher level of NUCB2/NESF-1in IDC cancer cells was noted compared to mastopathy. Also, the level of NUCB2 expression in the cytoplasm of IDC cells was shown to decrease with the increasing degree of tumor malignancy (G). A statistically significant higher NUCB2 expression was observed in tumors with estrogen receptor (ER)-positive and progesterone receptor (PR) positive phenotypes compared to estrogen receptor-negative and progesterone receptor-negative cases. A significant positive correlation between NUCBE/NESF-1 expression and expression of ER and PR ( p<0.001) was also observed. Moreover, a significantly higher expression was shown in ER(+) and PR(+) MCF-7 and T47D cell lines compared to triple negative MDA-MB-231 and normal human breast epithelial cells hTERT-HME1. Moreover, the analysis of five-year survival rate indicated that a positive NUCB2/NESF-1 expression (IRS ≥6) in tumor cells was also associated with longer patient survival (p=0.0186). Conclusions: The results suggest that NUCB2/NESF1 may play an important role in malignant transformation and may be a positive prognostic factor in invasive ductal breast carcinoma.
Point 3: Define in the text what is masopathy and why it is used as acontrol instead of normal
breast tissue
Response 3: Mastopathy is a benign, hormone-dependent change in the glandular tissue in the breast. The term “mastopathy” covers different benign changes in the mammary glands, such as nodules, swelling or cysts. Mastopathy is accepted as control in breast cancer studies (i.e. Gomulkiewicz A, Jablonska K, Pula B, Grzegrzolka J, Borska S, Podhorska-Okolow M, Wojnar A, Rys J, Ambicka A, Ugorski M, Zabel M, Dziegiel P. Expression of metallothionein 3 in ductal breast cancer. Int J Oncol. 2016 Dec;49(6):2487-2497. doi: 10.3892/ijo.2016.3759. Epub 2016 Nov 4. PMID: 27840910) because of difficulties related to obtaining normal breast tissue for research.
Line: 108
Point 4: What does mean "status" when talking about ER or NUCB2 in lines117-118? It that expression, activation, interaction? Authors should change the word "status" with more specific terms.
Response 4: ”ER status” is related to the expression of estrogen receptor in breast cancer cells and can indicate “ER-positive” cancer cells or “ER-negative” cancer cells. “ER-positive” means that the cells express the estrogen receptor on their surface and they grow in response to the hormone estrogen. Tumors that are ER-positive are much more likely to respond to hormone therapy compared to tumors that are ER-negative. The authors explained what the ER status is in the revised manuscript.
Lines: 120-121
Point 5: Lines 125-126, what does mean MCF-7 did not show invasion ability? Is that result described in control MCF-7 cells or MCF-7 cells with NUCB2 knockdown? Also, the lines above describe that removal of NUCB2 has an effect on invasion in MCF-7 cells, this an opposite result. Authors should better discuss this part of the manuscript.
Response 5: In 2013, Gest et al. evaluated breast cancer aggressiveness with MCF-7 cell line. Their conclusion was that the MCF-7 cells did not migrate or invade. Bearing in mind the above, MCF-7 cells seem to be a controversial model to investigate the role of NUCB2 in breast cancer invasion. We provided more details about this research in the revised manuscript.
Lines: 130-133
Point 6: What is exactly the expected subcellular location of NUCB2? Cytoplasm, membrane or nucleus? And how does the function of NUCB2 change depending on its subcellular location? This must be added in the text under a specific subheading (for example: "subcellular localization of NUCB2 and its related functions"). In addition, does the location and function change depending on the isoforms, nefstatin-1, 2 etc?
Response 6: We are afraid that there is too little information on the function of NUCB2 change depending on its subcellular location. We followed the Reviewer’s suggestion and placed the relevant information in the Discussion section.
Lines: 405-426
Point 7: Authors cited that NUCB2 as expressed in the pancreas. Pancreatic cancer is an important disease, authors should add a paragraph regarding the role of NUCB2 in pancreatic cancer
Response 7: There have been no studies on NUCB2 expression or the role in pancreatic cancer in the literature yet.
Point 8: Lines 180-186 cells were treated with recombinant nefstatin-1. How extracellular nefstatin-1 acts on the cells? Is there any specific known receptors
Response 8: Despite the recent advances in research on NUCB2/nesfatin-1, a putative receptor has not been identified yet. However, there are some observations which suggest that neuronal effects of nesfatin-1 are mediated via a G-protein coupled receptor (GPCR 3,6 and/or 12) [1]. Moreover, some studies indicated a connection between nesfatin-1 and the ghrelin receptor (growth hormone secretagogue receptor)[2]. Dore et al. suggested that nesfatin-1 signaling might not be directly linked to a putative NUCB2/nesfatin-1 receptor. They indicated that the nesfatin-1-induced activation of a cAMP response element (Cre)-reporter in a transfected neuroblastoma cell line and the suppression of cardiac L-type Ca2+ channels were attenuated by the melanocortin receptor MC3/4 antagonist SHU9119. However, it is known that nesfatin-1 does not directly act on the MC3/4. They concluded that some observations are not signaling events of a putative NUCB2/nesfatin-1 receptor but rather result from intracellular signaling cascades of receptors downstream the alleged NUCB2/ nesfatin-1 receptor, e.g. the MC3/4 [3]. To identify nesfatin-1 specific receptor and its mechanism of action, further studies are warranted. Identification, localization and modulation of the NESF-1 receptor will be the key for better understanding of nesfatin-1 signaling pathways.
Point 9: Line 196 what does biochemical recurrence-free survival stands for? This should be defined.
Response 9: Biochemical disease-free survival is the survival time of patients with cancer (e.g., prostate cancer) during which a biochemical marker (e.g., PSA) does not increase or increases slightly. The authors included this information in the revised manuscript.
Lines: 207-208
Point 10: Lines 233-234, what does G1, G2 and G3 stand for? Are those different stages of endometrial cancer?
Response 10: G1, G2 and G3 are tumor grades. Grading is the characteristic of a tumor based on what abnormal tumor cells and the tumor tissue look like under a microscope in comparison to normal controls. If the cells of the tumor and the tumor tissue are similar to normal cells and tissue, the tumor is known as “well-differentiated” (G1). These tumors grow and spread at a slower rate than tumors that are described as “moderately differentiated” (G2) or “poorly differentiated” (G3) which have abnormal-looking cells and do not have normal tissue structures. The grading is used to characterize most cancers, not only endometrial carcinoma. However, the factors used to determine tumor grade can vary between different types of cancer.
Point 11: Line 369, authors said that NUCB2 exert both pro-oncogenic and tumor suppressor roles. Surprisingly, the review cited only the pro-oncogenic roles of NUCB2, authors must add significant studies reporting a tumor suppressor role for NUCB2 in cancer. This information must be added to all cancer types cited in the review
Response 11: There is no information on a suppressive role of nesfatin-1 in cancer tissues. This is due to the fact that nesfatin-1 is a peptide encoded by nucleobindin-2 (NUCB2). Nesfatin-1 and nucleobindin-2 are colocalized in each tissue. There are no assays that could distinguish between proteolytically cleaved nesfatin-1 and full-length NUCB2. Only in some studies, Western blots were performed to distinguish between NUCB2 and nesfatin-1 based on the molecular weight. Consequently, in the present review, we refer to the analyte as NUCB2/nesfatin-1. In general, NUCB2 has a pro-oncogenic role in cancer tissues, but there are some in vitro studies which indicated recombinant nesfatin-1 as a tumor suppressor (i.e. in adrenocortical and ovarian cancer cells) which inhibited cell proliferation and increased apoptosis. The expression and the role of NUCB2/NESF-1 in adrenocortical and ovarian tumors has not been evaluated yet. The NUCB2 protein has several functional domains and proteolytic derivatives (i.e. nesfatin-1, nesftain-2, nesftain-3). We suspect that its role may be different in various tissues and cell types. To summarize, we concluded that NUCB2/nesftain-1 has both pro-oncogenic and tumor suppressor roles but further studies are warranted to assess the role of NUCB2/NESF-1 in cancer.
Thank you very much for your attention,
Sincerely Yours
Alicja Kmiecik
References:
- Brailoiu, G.C.; Dun, S.L.; Brailoiu, E.; Inan, S.; Yang, J.; Jaw, K.C.; Dun, N.J. Nesfatin-1: Distribution and interaction with a G protein-coupled receptor in the rat brain. Endocrinology 2007, 148, 5088–5094, doi:10.1210/en.2007-0701.
- Fan, X.T.; Tian, Z.; Li, S.Z.; Zhai, T.; Liu, J.L.; Wang, R.; Zhang, C.S.; Wang, L.X.; Yuan, J.H.; Zhou, Y.; et al. Ghrelin receptor is required for the effect of nesfatin-1 on glucose metabolism. Frontiers in Endocrinology 2018, 9, 1–13, doi:10.3389/fendo.2018.00633.
- Dore, R.; Levata, L.; Lehnert, H.; Schulz, C. Nesfatin-1: Functions and physiology of a novel regulatory peptide. Journal of Endocrinology 2017, 232, R45–R65, doi:10.1530/JOE-16-0361.
Round 2
Reviewer 2 Report
Thanks for the author’s effort revising the manuscript, the writing and formatting improved, I appreciate it. However, I’m still not confident in the functional importance of Nucleobindin-2/Nesfatin-1 in cancer, due to the lack of direct and mechanistic link of Nucleobindin-2/Nesfatin-1 and tumorigenesis. I’m afraid I still can’t recommend further consideration of this manuscript by IJMS for publication. Transferring to a more clinical focused journal may be an alternative choice.
Reviewer 3 Report
I am not convinced with the changes made by the authors. Their paper is only listing some clinical and experimental findings. The discussion that they added is still not enough. Overall, the review cannot be read smoothly.
A review paper is expected to focus on an exciting area to describe research findings, but this should be done by providing a dynamic discussion rather than just listing the findings. Nice and explanatory illustrations are also very helpful to catch the readers' attentions.
I encourage the authors to take time and significantly rewrite their paper. They should think about changing the whole structure and text to provide a more meaningful message for readers.
For these reasons and in order to maintain a policy of scientific rigor for IJMS, I recommend to reject the paper.